# The Methyltransferase Region of Vesicular Stomatitis Virus L Polymerase Is a Target Site for Functional Intramolecular Insertion

**DOI:** 10.3390/v11110989

**Published:** 2019-10-26

**Authors:** Emmanuel Heilmann, Janine Kimpel, Stephan Geley, Andreas Naschberger, Carles Urbiola, Tobias Nolden, Dorotheé von Laer, Guido Wollmann

**Affiliations:** 1Division of Virology, Medical University of Innsbruck, 6020 Innsbruck, Austria; emmanuel.heilmann@i-med.ac.at (E.H.); Janine.Kimpel@i-med.ac.at (J.K.); carles.urbiola@i-med.ac.at (C.U.); 2Christian Doppler Laboratory for Viral Immunotherapy of Cancer, Medical University of Innsbruck, 6020 Innsbruck, Austria; 3Division of Molecular Pathophysiology, Medical University of Innsbruck, 6020 Innsbruck, Austria; stephan.geley@i-med.ac.at; 4Division of Genetic Epidemiology, Medical University of Innsbruck, 6020 Innsbruck, Austria; andreas.naschberger@i-med.ac.at; 5ViraTherapeutics GmbH, 6020 Innsbruck, Austria; tobias.nolden@boehringer-ingelheim.com

**Keywords:** vesicular stomatitis virus, VSV, polymerase, insertion, methyltransferase

## Abstract

The L-protein of vesicular stomatitis virus (VSV) is a single-chain multi-domain RNA-dependent RNA polymerase. Previously reported attempts of intramolecular insertions of fluorescent proteins into the L-protein resulted in temperature-sensitive and highly attenuated polymerase activity. Here, we describe a novel insertion site that was selected based on in silico prediction. Of five preselected locations, insertion of the fluorescent protein mCherry in the VSV polymerase between amino acids 1620 and 1621 preserved polymerase function even after extended passaging and showed only mild attenuation compared to wildtype VSV polymerase. High magnification fluorescence imaging revealed a corpuscular cytosolic pattern for the L-protein. To confirm that the insertion site tolerates inclusion of proteins others than mCherry, we cloned mWasabi into the same position in L, generating a VSV-LmWasabi, which was also functional. We also generated a functional dual-color-dual-insertion VSV construct with intramolecularly labeled P and L-proteins. Together, our data present an approach to tag VSV polymerase intramolecularly without perturbing enzymatic activity. This L fusion protein might enable future tracing studies to monitor intracellular location of the VSV transcription and replication machinery in real-time life-imaging studies.

## 1. Introduction

Vesicular stomatitis virus (VSV) is a prototypical member of the family *Rhabdoviridae* and is widely studied as a model for viruses with nonsegmented negative-sense RNA genomes [1]. Its genome encodes five proteins in the following order from 3′ to 5′: nucleoprotein (N), phosphoprotein (P), matrix protein (M), glycoprotein (G), and polymerase (L). The viral genes are transcribed sequentially by the VSV polymerase, leading to a protein gradient from N to L [2,3]. The VSV L-protein is a single-chain multi-domain RNA-dependent RNA polymerase, which also catalyzes mRNA 5′-capping, cap methylation, and mRNA 3′ polyadenylation [4]. It is responsible for genome replication as well as mRNA transcription. Recently, the structure of the L-protein was revealed using cryo-electron microscopy. A structural organization of five distinct domains and two linker regions was described: an RNA-dependent RNA polymerase spanning amino acid positions 35–865; a capping domain at positions 866–1334; a linker 1 at positions 1335–1357; a connector domain at positions 1358–1557; a linker 2 at 1558–1597; a methyl-transferase at positions 1598–1892; and a C-terminal domain at 1893–2109 [5]. Replication occurs via a tripartite replicase complex formed by the N, P, and L-proteins, whereas the transcription complex is formed by P and L without N but rather with support from cellular proteins [6].

Stable intramolecular tagging with fluorescent proteins can support the study of viral protein function and was previously reported for two VSV proteins. The P-protein was shown to tolerate a green fluorescent protein (GFP) insertion at amino acid position 196 in its so-called hinge region [7] without significantly impairing its function. It was also demonstrated that the M-protein remains functional with an insertion of either eGFP or mCherry at amino acid position 37 [8]. Both VSV variants displayed only moderate attenuation. In contrast, a VSV construct with a G-protein c-terminally fused with GFP was replication competent and genetically stable only in the presence of unmodified G-protein either provided in trans or as an additional copy in the virus genome [9]. Previous attempts of inserting a fluorescent protein into the L-protein of VSV in the context of a fully replication-competent virus, however, were unsuccessful so far [10,11]. C- and N-terminal fusion proteins of the L-protein have also not been described thus far. Consequently, fluorescently tagging of the L-protein of VSV for life imaging and tracing for instance has remained elusive. Based on sequence comparisons of related Mononegavirales viruses from the genus morbillivirus for which successful insertions of fluorescent proteins have been described, it was attempted to generate an L-protein-eGFP variant with an insert site at aa1595. Though such recombinants could be rescued, replicative activity was limited by temperature sensitivity and viruses could not propagate at 37 °C beyond a few initial rounds of replication. Other attempted sites of insertion (1318, 1374, 1472, 1522, and 1577) resulted in polymerases without significant activity [10,11].

In this study, we used in silico prediction tools guided by the previously published L-protein structure to identify five potential in-frame insertion sites for mCherry. Locations at the surface and within flexible loops were factored into the selection process. Using a mini-genome assay, two out of the five selected insertion sites showed intact polymerase activity. For both L-protein variants, we generated full-length VSV plasmids of which one variant—VSV-L-MT1620-mCherry—yielded a replication-competent virus. Our data demonstrate, for the first time, the possibility to tag VSV polymerase intramolecularly without severely interfering with its enzymatic activity. Such a tool could potentially facilitate real-time tracing and kinetic studies of the VSV transcription and replication machinery in future studies.

## 2. Materials and Methods

### 2.1. Structure Visualization and Molecular Modeling

All structures were analyzed using Coot 0.8.7.1 [12] and UCSF (University of California, San Francisco) Chimera 1.12 [13]. Images of molecular structures were generated with UCSF Chimera 1.12. A VSV-L-MT1620-mCherry model was generated as follows: VSV L-protein (protein data bank (PDB) accession code 5a22) and mCherry (PDB accession code 2h5q) were docked with ZDock server [14]. VSV L-protein was defined as the reference structure to which unrestrained mCherry was docked in rigid body mode. One of the top hits was chosen because N- and C-termini of mCherry were located nearby the MT1620 insertion site. Subsequently, FiberDock [15] was used for flexible refinement of the rigid-body protein–protein docking solution. The (GGSG)_3_-Linkers were manually introduced in Coot 0.8.7.1 and modeled using ModLoop [16].

### 2.2. Viruses and Cells

Generation of VSV and VSV-GFP was described previously [17,18]. VSV-GFP-ΔL was cloned and produced analogous to VSV∆L-DsRed described previously [19]. BHK-21 cells (American Type Culture Collection, Manassas, VA, USA) were cultured in Glasgow minimum essential medium (GMEM) (Lonza) supplemented with 10% fetal calf serum (FCS; Thermo Fisher Scientific, Vienna, Austria), 5% tryptose phosphate broth (Gibco, Carlsbad, CA. USA), 100 units/mL penicillin (Gibco), and 0.1 mg/mL streptomycin (Gibco). 3T3 cells (kind gift from Tim Fenton, UCL London, UK), 293T (American Type Culture Collection, Manassas, VA, USA), A549 (DSMZ, Germany), HEp-2 (CLS, Eppelheim, Germany), and 293-VSV (293 expressing N, P-GFP, and L of VSV; [20]) were cultured in Dulbecco’s Modified Eagle Medium (DMEM) (Lonza) supplemented with 10% FCS (Invitrogen), 1% P/S (PAA Laboratories), 2% glutamine (PAA Laboratories), 1× sodium pyruvate (Gibco), and 1× nonessential amino acids (Gibco).

### 2.3. Plasmid Construction

VSV L-expression plasmid insertions were introduced by three-fragment Gibson assembly [21]. The mCherry insert (fragment 1) was framed with a GGSGGGSGGGSG linker ((GGSG)_3_) sequence with the forward primer mCherry-GGSG for (5′-GGCGGAAGCGGCGGAGGGAGCGGGGGCGGGAGCGGAATGGTGAGCAAGGGCGAGG-3′) as well as with the reverse primer mCherry-GGSG rev (5′-GCCGGATCCACCGCCTGAGCCGCCTCCGGACCCTCCCTTGTACAGCTCGTCCATG-3′). Two overlapping PCR products (fragment 2 and 3) were generated from the L-protein expression vector pCI-Neo-L that contained complementary sequences to the (GGSG)_3_-linker sequences of fragment 1. Fragments 2 and 3 contained overhangs to the (GGSG)_3_ linker at the 5′ end with primer n*-insertGGSG rev and 3′ end with primer n*insertGGSG for, respectively.

Splitting of the expression vector in fragments 2 and 3 was performed to increase PCR product yield of an otherwise unfavourably large (11,767 bp) vector. Depending on the insertion site, the primers contained complementary sequences to the L-protein of 19–22 bp and a 21 bp part that was complementary to the GGSG-linkers (see Table 1).

VSV L virus insertions were introduced by a four-fragment Gibson assembly. The larger part of the vector was provided by restriction enzyme digestion with enzymes SfoI and FseI of either pVSV-gfp for VSV-GFP-L-mCherry or pVSV-XN2-1 for VSV-L-mCherry, VSV-L-mWasabi, and VSV-L-GFP vectors. This fragment is referred to as fragment 4. The mCherry/mWasabi insert (fragment 1) was framed with two (GGSG)_3_ linker sequences with forward primer mCherry/mWasabi-GGSG for (5′-GGCGGAAGCGGCGGAGGGAGCGGGGGCGGGAGCGGAATGGTGAGCAAGGGCGAGG-3′) and reverse primer mCherry/mWasabi-GGSG rev (5′-GCCGGATCCACCGCCTGAGCCGCCTCCGGACCCTCCCTTGTACAGCTCGTCCATG-3′). L-protein sequences surrounding fragment 1, hereafter referred to as fragments 2 and 3, received overhangs to the (GGSG)_3_ linker at the 5′ end with primer n*-insertGGSG rev and 3′ end with primer n*-insertGGSG for, respectively. GFP was PCR amplified with primers GFP for and rev (Table 1). VSV-L-CD1595-GFP [11] fragments 2 and 3 were generated with primers CD1595insertGFP for and rev (Table 1). These fragments vary slightly in length in every construct depending on the insertion site. Fragment 2 received an overhang to fragment 4 with forward primer 49bp-before-FseI (5′-GCTGCCAAGTAATACACCGG-3′). Fragment 3 received an overhang to fragment 4 with reverse primer 50bp-after-SfoI (5′-TTTATCTCCTCCTAAAGTTTC-3′) (*position of respective insertion site; see Table 2). Inserts were PCR size and sequence confirmed.

VSV-P-mCherry-L-mWasabi was generated by the insertion of mCherry at aa position 196 [7] in VSV-L-mWasabi using primers 196-GGSG-P for and rev to generate fragments between the nearest restriction enzyme sites for Bst1107l and XbaI with overlaps to VSV-L-mWasabi cut with Bst1107l and XbaI and GGSG linkers. The same mCherry fragment framed with (GGSG)_3_ was used as in L-insertion viruses. Additionally, mutations observed in one VSV-L-mWasabi variant near the sites of insertion (K1402R and M1936I) were introduced by Gibson assembly site-directed mutagenesis. VSV-P-mWasabi-L-mCherry was generated analogously to VSV-P-mCherry-L-mWasabi with the starting vector VSV-L-mCherry instead of VSV-L-mWasabi, being digested with Bst1107l and XbaI. (GGSG)_3_ framing primers for mCherry could be used for mWasabi as well because the N- and C-terminal sequences of mCherry and mWasabi are identical.

To exclude random insertions of mCherry in other part of the VSV genome, additional insert control primers more distant to mCherry were used: for. 3′-CATGCCGAGGACAGTTCTCTAT-5′ and rev. 3′-ATTTCCTCCGACTCAAAGCAG-5′.

VSV, VSV-GFP, VSV-L-mCherry, and VSV-GFP-L-mCherry regions framing the mCherry insert were compared in agarose gel electrophoresis. VSV, VSV-GFP: 1114 bp; VSV-L-mCherry, VSV-GFP-L-mCherry: 1893 bp.

### 2.4. Mini-Genome Assay

Transfection of L-mCherry expression plasmids was performed with a TransIT^®^-LT1 transfection kit from Mirus in 293T cells. Plasmid DNA and transfection reagent amounts were chosen according to manufactures’ recommendations for 24-well plates, in which 2.7 × 10^5^ 293T cells per well were seeded one day before transfection. P-expression plasmids were co-transfected with L-mCherry expression plasmids. Twenty-four h after transfection, 293T cells were infected with VSV-GFP-ΔL at a multiplicity of infection (MOI) of 10. Images were acquired 48 h after infection.

### 2.5. Virus Recovery

VSV-L-MT1620-mCherry, VSV-GFP-L-MT1620-mCherry, VSV-L- MT1620-mWasabi, VSV-L-CD1595-GFP, and VSV-P-mCherry-L-mWasabi were rescued in 293T cells by CaPO_4_ transfection of whole-genome VSV plasmids with L-mCherry insertions together with N-, P-, M-, G-, and L-expression plasmids as helper plasmids as described previously [22]. M and G-proteins as helper plasmids are optional in the recovery of VSV, chosen here as a precaution to support the rescue of potentially attenuated virus variants. After the rescue, viruses were passaged on 293-VSV cells and plaque purified twice on BHK-21 cells. VSV-L-CD1595-GFP stocks were produced at the permissive temperature of 33 °C [11].

### 2.6. Immunoblotting

BHK-21 cells were infected with VSV, VSV-GFP, VSV-L-MT1620-mCherry, or VSV-GPF-L-MT1620-mCherry at an MOI of 5, and cell lysates were prepared 8 h later. Uninfected BHK-21 cells were used as a control. Cells were lysed in ice-cold cell lysis buffer (50 mmol/liter HEPES, pH 7.5; 150 mmol/liter NaCl; 1% Triton X-100; 2% aprotinin; 2 mmol/liter EDTA, pH 8.0; 50 mmol/liter sodium fluoride; 10 mmol/liter sodium pyrophosphate; 10% glycerol; 1 mmol/liter sodium vanadate; and 2 mmol/liter Pefabloc SC) for 30 min. To dispose of cellular debris, cell lysates were centrifuged at 13,000 rpm for 10 min. Supernatants containing proteins were stored at −80 °C.

SDS-PAGE of protein lysates was performed under reducing conditions on a 12% polyacrylamide gel. For comparison of VSV, VSV-GFP, VSV-L-MT1620-mCherry, and VSV-GFP-L-MT1620-mCherry, the 8-h time point lysates were used. Proteins were transferred to 0.45-µm nitrocellulose membranes (Whatman, Dassel, Germany) by using a tank blotting system. The blotting time was 90 min. The membrane was blocked overnight with 1 × PBS containing 5% skim milk and 0.1% Tween 20 (PBSTM) and incubated for 3 h at room temperature with an mCherry-specific rabbit monoclonal antibody diluted 1:1000 in PBSTM. The antibody was raised in house against recombinant mCherry and affinity purified. After washing, a peroxidase-conjugated rabbit IgG-specific antibody from goat (Invitrogen, Carlsbad, CA), diluted 1:5000 in PBSTM was added and the blot was incubated for another hour. After washing, blots were developed with enhanced chemiluminescence (ECL). After the first detection, the same blot was reused to stain for loading control. Actin was stained with a β-actin specific monoclonal antibody from mouse (A2228; Sigma, Munich, Germany) diluted 1:5000 in PBSTM, and a secondary horseradish peroxidase-conjugated mouse IgG-specific antibody from goat was used.

### 2.7. Virus Sequencing

Genomic viral RNA of viruses generated in this study was purified with PeqGOLD Viral RNA kit (Peqlab), and cDNA synthesis was performed with RevertAid First Strand cDNA Synthesis Kit (Thermo Fisher Scientific) according to manufacturer’s recommendations. Ten overlapping PCR products per virus genome were generated. Subsequently, PCRs were column purified and Sanger sequenced by Microsynth AG (Balgach, Switzerland) with up to 40 different primers, i.e., sequencing reactions, depending on amount of inserts (1 or 2) and sequence quality, to obtain whole genome sequences.

### 2.8. Plaque Assay

Five-fold serial dilutions of virus stocks were prepared and used to infect BHK-21 cells at 60% confluency in 6-well plates. One hour after infection, cells were washed with PBS and covered with a 2.5% plaque agarose/GMEM mixture. After 24 h at 37 °C, plaques were visualized by fluorescence microscopy.

### 2.9. Single Step Growth Curve and TCID_50_ Assay

In 12-well plates, 10^5^ cells per well were seeded one day before infection. BHK-21, 293T, HEp-2, A549, and 3T3 cells were infected in duplicates with a multiplicity of infection (MOI) of 3 of the respective VSV variants at either 33 °C or 37 °C, respectively. One hour after infection, the medium was removed, cells were washed twice with PBS, and fresh medium was added. Supernatant was collected at the indicated time points and stored at −80 °C until further analysis. For quantification, the 50% tissue culture infective dose (TCID_50_) assay was performed as described previously [23]. In short, 100 µL of serial dilutions of virus were added in octuplicates to 10^3^ BHK-21 cells seeded in a 96-well plate. Six days after infection at 37 °C (or 33 °C for VSV-L-CD1595-GFP), the TCID_50_ were read out and titers were calculated according to the Kaerber method [24].

### 2.10. Interferon (IFN) Sensitivity and Cell Viability Assay

Virus cell killing was assessed in an Interferon-response assay, in which interferon (IFN)-competent BHK-21 cells were treated with increasing amounts (10, 100, 500, and 1000 U/mL) of recombinant universal type I IFN (PBL assay science, Piscataway Township, NJ) and infected with MOIs 0.1, 1, and 10. Cells were seeded at 10^4^ one day before IFN treatment. IFN treatment was performed 16 h before infection. Seventy-two h at 37° post infection, thiazolyl blue tetrazolium bromide (MTT) was added for 4 h. Cells were then dissolved in 0.1 M HCl with 1% SDS for another 4 h. Colorimetric changes were measured at 540 nm.

### 2.11. Virus Passage

VSV-GFP-L-MT1620-mCherry, VSV-L-MT1620-mCherry, and VSV-L-CD1595-GFP were serial-passaged for 10 times at 37 °C or until no more virus could be detected. VSV-L-CD1595-GFP passage was initiated at the permissive temperature of 33 °C followed by 37 °C conditions. For each passage, 2 µL of supernatant were transferred to 3 × 10^5^ cells in 1 mL medium per 12-well dish well after 24 h of incubation. Titers were determined with TCID_50_ at 37 °C for VSV-L-MT1620-mCherry passages and at 33 °C for VSV-L-CD1595-GFP.

### 2.12. Fluorescence Microscopy and Time-Lapse Recordings

One day before infection, 2 × 10^5^ BHK-21 cells were seeded in polylysine-coated glass bottom dishes (ibidi GmbH, Gräfelfing, Germany). Nuclei were stained with 0.5 µM Sir-DNA (tebu-Bio GmbH, Offenbach, Germany) 30 min prior to infection, and P glycoproteins were inhibited with 5 µM verapamil for better Sir-DNA staining. BHK-21 cells were infected with an MOI of 10. Single images were acquired 6–10 h after infection at 37 °C using a 63X/NA1.4 objective on an automated live cell imaging Zeiss Axiovert 200M microscopy equipped with a Sola light engine LED light source (Lumencor, Visitron Systems GmbH, Puchheim, Germany) and a pco.edge 4.2 scMOS camera (PCO AG, Kelhaim, Germany) and controlled by VisiView software (Visitron). Exposure times were 200 ms for mCherry/mWasabi, 50 ms for Sir-DNA, and 10 ms for phase contrast. For time-lapse recordings, images were taken for 8 h every 10 min from 10 different positions.

## 3. Results

### 3.1. In Silico Prediction of Potential mCherry Insertion Sites

Using in silico prediction and recent information on the 3D structure of the L-protein, we focused our search for potential insertion sites on the connector domain (CD; 1358–1557) and the methyltransferase domain (MT; 1598–1892) (Figure 1A,B). We narrowed the region for possible insertions sites in the L-protein to the three globular domains, which were described to fold independently [25], therefore possibly being more permissive for insertion. Additional factors were to avoid the core catalytic domains as well as structured surfaces, which might interact with other VSV proteins. We also avoided insertion sites within alpha-helices and beta-sheets to conserve structural integrity of the protein. Finally, we selected regions located at the surface and in flexible loops, which should minimize the possibility of steric clashes. This led to the identification of the following candidate insertion sites: CD1506, CD1537, MT1603, MT1620, and MT1889 (Figure 1A,C).

We chose the fluorescent protein mCherry as an insert protein due to its activity in monomeric form without the need to dimerize. We also theorized that intramolecular insertions would benefit from adjustable connections and, hence, included two (GGSG)_3_-linkers flanking the mCherry sequence within the construct. A projected model of an mCherry inserted into position MT1620 of the VSV L-protein is shown in Figure 1D.

### 3.2. VSV-GFP-∆L Screening Points towards Two Promising Insertion Sites

To accelerate the screening of the candidate insertions, we chose to apply a mini-genome approach. L-mCherry expression vector plasmids with insertions at CD1506, CD1537, MT1603, MT1620, and MT1889 were generated. After transfection of these five constructs in 293T cells, infection with a propagation-incompetent VSV-GFP-∆L virus, which lacks the gene for the viral polymerase L and codes for eGFP as reporter, was performed. In this screening, all sites showed mCherry signal to various extent—indicative for functional expression of the mCherry within the L-protein sequence. However, only two sites (CD1506 and MT1620) showed eGFP signal, indicating transcriptional activity of the respective L-mCherry fusion protein (Figure 2A). Thus, every insertion site allowed correct mCherry folding, although with varying efficiency, but only two insertions retain polymerase activity.

### 3.3. MT1620 Insertion is Compatible with VSV Replication

Next, we cloned the mCherry gene into the full-length VSV genome at L-protein sites CD1506 and MT1620. Insertions were performed into two VSV backbones, one with eGFP as an additional reporter at fifth position in the genome, the other without eGFP (see schemes above the photomicrographs in Figure 2B). Generation of VSV-L-MT1620-mCherry and VSV-GFP-L-MT1620-mCherry yielded replication competent viruses, which was detected by the cytopathic effect (CPE) and fluorescent signal (Figure 2B). In contrast, the attempt to rescue VSV-L-CD1506-mCherry and VSV-GFP-L-CD1506-mCherry failed. Expectedly, VSV-GFP-L-MT1620-mCherry showed fluorescent signals in both the fluorescein isothiocyanate (FITC) and tetramethylrhodamine (TRITC) channels and VSV-L-MT1620-mCherry showed fluorescent signals only in the TRITC channel (Figure 2B). To address whether the insertion site only tolerates mCherry or other reporter genes also, we cloned and rescued VSV-L-MT1620-mWasabi. The resulting recombinant construct showed green fluorescence and intact polymerase activity comparable to the mCherry counterpart. Notably, mWasabi fluorescence yield is stronger than mCherry’s [26], which results in earlier and stronger focal fluorescence signals.

To verify mCherry presence at the protein level, we performed immunoblots with an mCherry specific antibody. BHK-21 cells were infected with VSV, VSV-GFP, VSV-L-MT1620-mCherry, and VSV-GFP-L-MT1620-mCherry. As a positive control, a vector containing only mCherry was transfected in BHK-21 cells. mCherry inside L protein displayed a signal at high molecular weight (expected at 267 kDa), in accordance with the production of the L-mCherry fusion protein after viral infection (Figure 2C).

Taken together, these results show the successful insertion of mCherry at position MT1620, leading to a replication competent virus.

### 3.4. Viral Genome Sequencing Reveals Secondary Mutations

Upon whole genome sequencing of rescued VSV-L-insertion variants, we observed one to two secondary non-synonymous mutations in all viruses, which are located in proximity to the site of insertion (Appendix A). We did not find any silent mutations. The mutations in L-mWasabi viruses were located upstream (mWasabi 1: K1402R; mWasabi 2: R1410T) and downstream (mWasabi 1: M1936I; mWasabi 2: I1899L) of mWasabi. VSV-L-mWasabi with mutations K1402R and M1936I was used for further experiments (including for generation of a double insertion virus described below). The mutations in L-mCherry viruses were located downstream of mCherry (VSV-GFP-L-mCherry (mCherry 1): C2098G, E2107A; VSV-L-mCherry (mCherry 2): N2109Y). We were not able to rescue viruses that did not harbor associated mutations in the L protein. Whether these mutations are conditional and required for proper polymerase function remains to be studied.

### 3.5. Insertion of mCherry at Position MT1620 Results in Temperature-Stable VSV Recombinant with Mildly Attenuated Virus Replication and Activity

We next compared MT1620 mCherry and the previously published temperature-sensitive CD1595 GFP insertion variants with wildtype L VSV counterparts for their replication kinetics on BHK and HEp-2 cells at 33 °C and 37 °C (Figure 3). VSV-GFP and VSV-L-MT1620-mCherry yielded similar titers in different cell lines both at 33 °C and 37 °C (Figure 3B). In contrast, VSV-L-CD1595-GFP yielded a wild-type comparable titer only at 33 °C compared to a titer reduction of about 2 logs at 37 °C in both cells tested, confirming previous reports on the temperature-labile characteristics of the L-CD1595-GFP mutant. To confirm that the temperature stable activity of the VSV-L-MT1620-mCherry can generalize to other cells, we performed single-step growth curves on human 293T and A549 cells and mouse 3T3 cells. No differences in titer were observed between incubations at 33 °C and 37 °C, and titer differences between the virus variants were only marginal (Figure 3C). We next compared viral activity in a cell viability assay on BHK-21 cells under increasing doses of IFN type 1 (Figure 3D). IFN was added 16 h before infection. Compared to VSV and VSV-GFP, VSV-L-MT1620-mCherry showed only mildly reduced cytotoxic activity, and a high MOI could overcome the protective effects of IFN-1 on BHK cells. In comparison, cell killing was strongly reduced with VSV-L-CD1595-GFP with high MOIs unable to neutralize the IFN-1 effect. Together, our data support that insertions of mCherry at L-protein position MT1620 are well tolerated, resulting in only mild attenuation compared to wild-type VSV in contrast to previously described insertions [10,11].

### 3.6. VSV L-MT1620-mCherry Insert Variants Are Genetically Stable

To assess genetic stability, VSV-GFP-L-MT1620-mCherry and VSV-L-MT1620-mCherry were passaged 10 times at 37 °C. Plaque assays were performed before and after passaging. At both endpoints, all of the 100 counted phase contrast plaques were positive for mCherry fluorescence, as shown in exemplary VSV-L-MT1620-mCherry plaques (Figure 4A). In addition, sequencing of both VSV-L-MT1620-mCherry and VSV-GFP-L-MT1620-mCherry confirmed genetic stability in the L-protein insertion region and the adjacent segments, where secondary mutations were observed initially in the course of a genome-wide sequencing (Appendix A). As passage of the L-CD1595 insertion variant had previously been reported to be abortive after 4 cycles [11], we compared viral titers in the course of sequential passage at 37 °C of VSV-L-MT1620-mCherry and VSV-L-CD1595-GFP (Figure 4B). Viral titers of the MT1620-mCherry insertion variant remained comparable with starting titer throughout the passages, unlike the CD1595-GFP mutant, which seized propagation after 3 passages.

### 3.7. High Magnification Fluorescence Microscopy Shows Viral Inclusion Bodies and Distinct Patterns of Localization of GFP, L-mCherry, and L-mWasabi

It has been reported that VSV forms so-called viral inclusion bodies (VIBs) during replication, which are known to contain concentrated levels of P and L protein [27]. To investigate VIB formation, we used high-resolution fluorescence time-lapse microscopy. We observed distinct cytosolic foci of mCherry fluorescence suggestive of VIBs, although no validation was performed (Figure 5A). Nuclear dye SiR-DNA (infrared channel) confirmed the cytosolic character of the fluorescence signal. We also compared eGFP and mCherry signals with the double fluorescent VSV-GFP-L-MT1620-mCherry. eGFP signal arose earlier in the infection and was diffuse, whereas mCherry signal was focal with a distinct nuclear sparing. VIB-like signals were detectable over a range of 4–16 h using time-lapse analysis (Figure 5B, Appendix A), though the distinct corpuscular signal disappeared during late-stage virus replication and mCherry signal became diffuse throughout the cell (not shown). To test whether position MT1620 of the L protein also tolerates insertion of a protein other than mCherry, we generated VSV-L-MT1620-mWasabi. This variant displayed similar growth characteristics as the mCherry insertion mutant (see below and Figure 6B). Of note, the mWasabi fluorescence signal was markedly more corpuscular than mCherry reporter (Figure 5C), presumably due to the multifold stronger brightness of mWasabi compared to mCherry [26].

### 3.8. A Dual Insertion VSV-P-mCherry-L-mWasabi Construct Displays Co-Localization of Fluorescently Tagged P and L Protein

In order to assess the feasibility of multi-intramolecular labeling within a single VSV variant, we generated dual fluorescence VSV-P-mWasabi-L-mCherry and VSV-P-mCherry-L-mWasabi combining the previously described VSV P-protein insertion site [7] with our L-MT1620 insertion site. Both viruses showed co-localizing fluorescence signals in FITC and TRICT channels, putatively representing the interaction of P-mCherry and L-mWasabi fusion protein interactions (Figure 6A). Both double insertion viruses produce significantly lower titers compared to parental VSV (up to 5 logs at 12–24 h), VSV-GFP, and single L-insertion VSV variants (about 2 logs at 12–24 h), as shown in a multistep (MOI 0.1) replication kinetic at 37 °C (Figure 6B).

## 4. Discussion

The VSV polymerase, which is referred to as L protein (for “large protein”), is a single chain multi-domain RNA-dependent RNA polymerase, which catalyzes genome replication as well as mRNA transcription, capping, and methylation. We present here a novel insertion site into which relatively large proteins such as mCherry or mWasabi can be introduced as a novel functional domain into the L protein without deleteriously impairing replication competence. However, L-insertion VSV variants were slightly attenuated relative to parental VSV.

This intramolecular tagging approach is a strategy to express additional genes from a viral genome. The more common approach is the insertion of an additional gene alongside an extra intergenic region to extend the natural transcription gradient of the virus to the added gene [28]. Respective to the transgene position, its quantity is lesser than the preceding and greater than the following gene. There are multiple rationales to introduce transgenes into VSV, e.g., to incorporate therapeutic genes, vaccine antigens, or marker genes to mark infected cells or viral proteins. Intramolecular insertions in the context of vaccine antigens can be advantageous because antigens would already be present in (and presented by) the virion. An advantage of intramolecular insertions in the context of tracing studies is that single VSV protein detection becomes possible. A disadvantage of intramolecular insertions is the possibility for attenuation, which appears to be associated with some intramolecular insertions compared to an insertion at additional intergenic regions [7,8].

Previous attempts to generate replication-competent VSV variants with an L-protein eGFP insertion resulted in viruses that were either unable to replicate or temperature sensitive [10,11]. These L-protein eGFP insertions were generated based on sequence similarities of VSV L with the polymerase of related Mononegavirales viruses, where insertions had been successful, e.g., measles virus (MV) [29], rinderpest virus (RPV) [30], and canine distemper virus (CDV) [31]. The structure of the VSV L protein was revealed after these initial insertion attempts [5], and some of the shortcomings of the previously studied insertion sites can now be understood in retrospect. For example, one published insertion site at aa1472 is found to be located in a groove between the connector domain (CD), methyltransferase (MT), and the C-terminal domain (CTD). Hence, the insertion of GFP resides in the hydrophobic core of the polymerase and would likely produce steric clashes [32]. As the published insertion at aa1595 is located almost at the end of a linker where P-protein is assumed to bind [5], the resulting variant showed some residual activity, as recently confirmed by another study [33]. On the other hand, some of the other described insertions in closer vicinity to the P-protein binding site may impair proper interaction. Finally, it could be hypothesized that the attempted insertion site aa1318, due to its proximity to a cysteine at position aa1302 could destabilize the loop between aa1302 and aa1318 and, therefore, interfere with structural integrity.

Here, we predicted potentially suitable insertions sites based on the published L structure. As shown in Figure 1, we specifically chose flexible surface-bearing loops, which we expected to allow heterologous protein insertions without disturbing the functional domains of the L protein, resulting in the selection of sites CD1506, CD1537, MT1603, MT1620, and MT1889. The viruses with the insertion at L-MT1620, despite minor attenuation, proved to be replication competent and were further characterized. This minor attenuation may be due to L-protein-mCherry fusion protein misfolding as well as disturbance of structural rearrangements in L, which occur during viral genome replication and mRNA transcription, capping, and methylation. However, we did not observe temperature sensitivity when comparing VSV-L-MT1620-mCherry with VSV-GFP at 33 °C and 37 °C in different cell lines. Notably, we performed replication kinetics on HEp2 cells, which have been shown to be nonpermissive for VSV variants with impaired methyltransferase activity [34]. BHK-21 cells have high intrinsic methylase activity that can rescue viruses with impaired methyltransferase activity [34]. Since we made our insertion in the methyltransferase domain and used BHK-21 cells in most of our experiments, HEp-2 cells were an important indicator for the possible disturbance of methyltransferase activity in our insertion viruses. We observed only marginal titer differences between 33 °C and 37 °C in HEp-2 cells, possibly indicating a slight methyltransferase activity reduction. When we compared VSV-L-MT1620-mCherry with previously described VSV-L-CD1595-GFP, we observed titer reductions up to 3 logs for the latter construct. We tested the integrity of mCherry insertions by several techniques. The region of the insert was sequenced. Second, we showed that mCherry appears at high molecular weight in immunoblotting, where one would also expect L protein. Third, in contrast to 5th-position GFP signals, which were diffuse, mCherry fluorescent signals were focal in high magnification microscopy imaging, which suggests co-localization with L-protein replication bodies. The phenomenon of VSV to form cytoplasmic inclusions has been described previously and was attributed to focal accumulations of the viral replication machinery [27,33]. The focal accumulations of L-mCherry or L-mWasabi proteins in VSV-L-MT1620-mCherry-infected cells are highly reminiscent of the replication bodies described for VSV and other viruses, such as rabies virus, where the phenomenon is termed Negri body-like structures (NBLs) [35].

In our time-lapse infection experiment, we saw that 5th-position GFP signal manifests prior to mCherry signal. The sequential gene expression of VSV is one of the fundamental principles of negative-strand RNA viruses of the order of mononegaviruses [36]. The sequential gene expression of VSV supports our data, since GFP is located at position 5 in our construct, followed by the L-protein-mCherry at position 6. This leads to larger amounts of GFP compared to mCherry and, consequently, an earlier fluorescence detection. However, the later appearance of the mCherry signal could also be due to its weaker fluorescent capacity [37] as well as the possible misfolding of some L-protein-mCherry fusion proteins.

RNA virus polymerases have high error rates mainly due to missing proofreading activity of RNA-virus polymerases [38,39]. It was described previously that mutations introduced in a flexible structural motif of the VSV N protein are complemented in cis by additional N mutations and in trans by mutations in the P and L proteins to restore activity of the N-P-L complex [40]. We found mutations in our viruses near the site of insertion. Viruses with the same inserts produced mutations in similar sequence areas. We, therefore, speculated cautiously that these cis-complementing mutations may be necessary for proper fusion protein function.

To assess genetic stability of our insert position in comparison with a previously reported position [10,11], we passaged VSV-GFP-L-MT1620-mCherry, VSV-L-MT1620-mCherry, and VSV-L-CD1595-GFP with low MOI at 37 °C. Virus plaques were checked for overlap of phase contrast plaques and mCherry fluorescence in VSV-L-MT1620-mCherry before and after passaging both L-mCherry insert viruses 10 times at 37 °C. After passaging, we found no plaques lacking mCherry fluorescence. As shown previously, mutations in GFP—resulting in changes of the proteins fluorescence spectrum—can be found after VSV-GFP passaging [41]. The preservation of red fluorescence in all observed plaques may indicate that mutations or deletions in mCherry could cause structural changes, impairing L-protein function and aborting propagation of mutated insert variants. We titrated VSV-L-MT1620-mCherry and VSV-L-CD1595-GFP passages and found that VSV-L-MT1620-mCherry retained high titers of 10^8^ TCID_50_ in each passage at 37 °C, other than VSV-L-CD1595-GFP, for which titers declined after only a few passages.

The VSV L-mCherry fusion protein presented here may enable the more detailed study of VSV replication, especially when combined with previously described P-GFP [7] and M-GFP [8] insertions. In a first proof-of-concept, we generated dual fluorescence viruses VSV-P-mWasabi-L-mCherry and VSV-P-mCherry-L-mWasabi. The brighter fluorescent protein mWasabi was used in the L protein to balance the expression gradient. Though attenuated in the replicative capacity compared to parental VSV, we observed co-localization of mCherry and mWasabi signals, which is indicative of the known interaction of P and L proteins. Functional characterizations, for example, with inhibitors of VSVs replication and transcription machinery, of this construct will be the subject of future studies.

## 5. Patents

Results from this study supported a recent patent application.

## Figures and Tables

**Figure 1 viruses-11-00989-f001:**
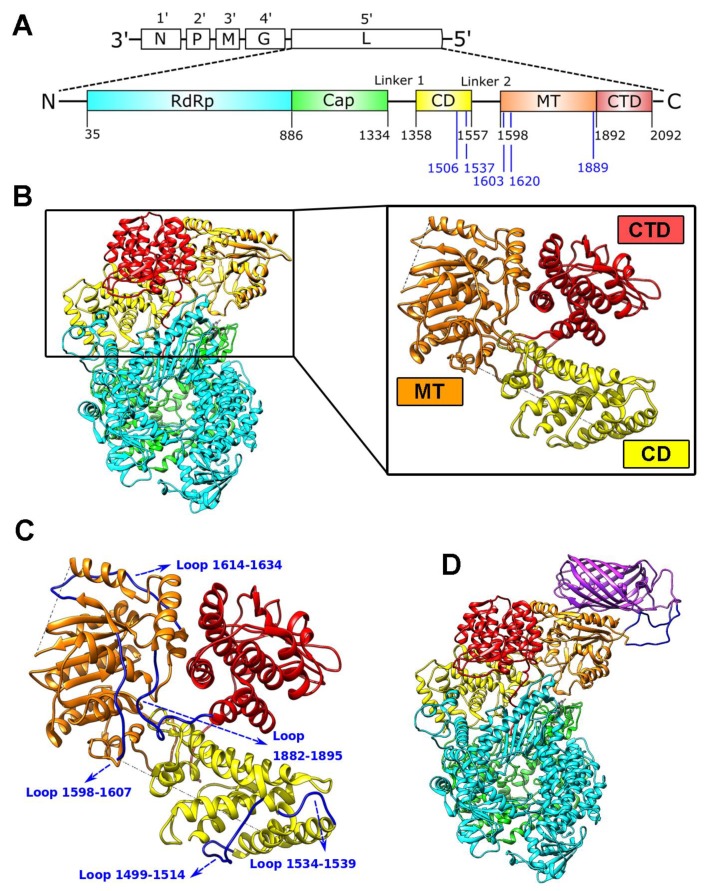
Domain organization, structure, and insertion sites in the VSV L-protein. (**A**) Schematic VSV genome organization showing the genes in 3′to 5′ directions and the VSV L-protein domain scheme with its corresponding domain borders labeled with black numbers [5]. Blue labels indicate candidate insertion sites tested herein. (**B**) VSV L-protein structure as determined by Liang et al. [5]:. Right panel visualizes the zoom to connector domain (CD; in yellow), methyltransferase domain (MT; in orange), and the C-terminal domain (CTD; in red). (**C**) Zoom on CD, MT, and CTD (colors as above) with loops indicated in blue that were chosen as insert site. (**D**) Molecular model of VSV L-protein with mCherry (depicted in purple) insertion at position MT1620.

**Figure 2 viruses-11-00989-f002:**
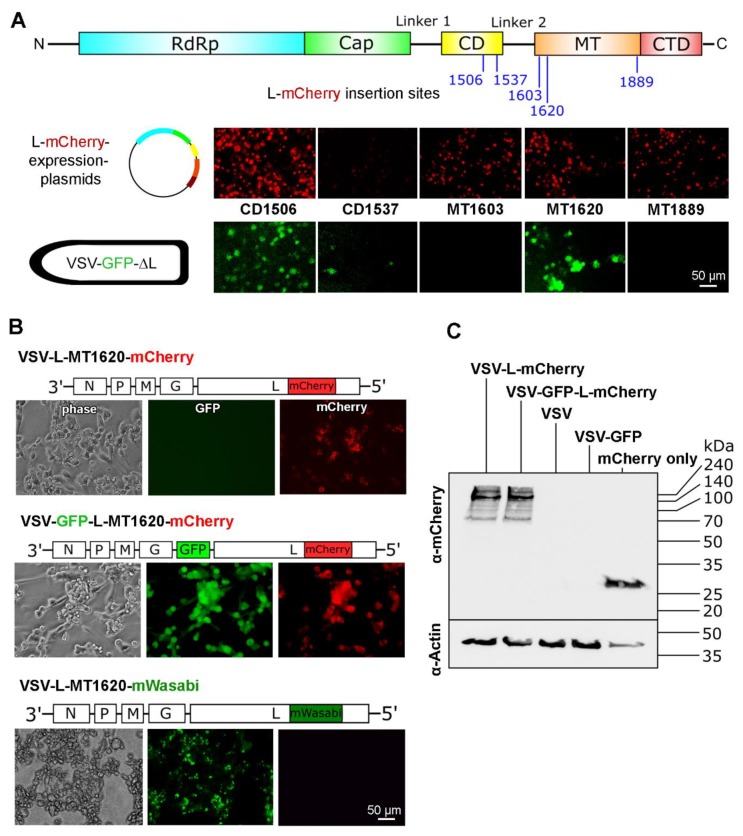
Insertion of mCherry at position MT1620 is compatible with VSV replication. (**A**) Top: VSV L-protein domain scheme with insertion sites. Middle: Images of 293T cells transfected with five different L-mCherry expression plasmids. The corresponding insertion sites are labeled with the domain abbreviation followed by amino acid number. Red indicates L-mCherry expression. Bottom: GFP signal after inoculation with VSV-GFP-ΔL at a multiplicity of infection (MOI) of 10 depicted from the same micrograph frames as above. Green fluorescence indicates functional L-mCherry fusion proteins and polymerase activity. Scale bar 50 µm. (**B**) Fluorescence and phase contrast images of VSV-L-mCherry and VSV-GFP-L-mCherry 24 h after infection of BHK-21 cells: Virus genome schemata are displayed above the fluorescence images. Scale bar 50 µm. (**C**) Immunoblot against mCherry under reducing conditions on 12% polyacrylamide gel: β-actin was used as loading control. VSV, VSV-GFP, VSV-L-mCherry, and VSV-GFP-L-mCherry infected BHK-21 cells 8 h after infection were used to prepare lysates.

**Figure 3 viruses-11-00989-f003:**
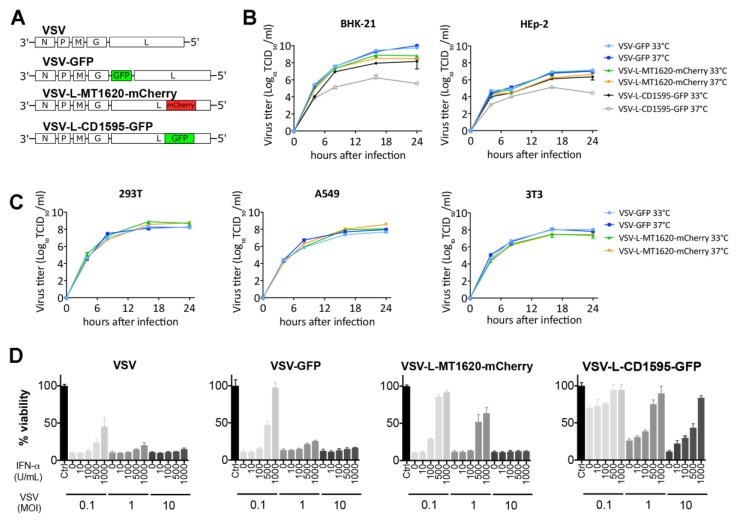
L-Insertion site MT1620 performed better than previously described CD1595. (**A**) Virus genome schemes of used viruses: VSV; VSV-GFP; VSV-L-MT1620-mCherry; and VSV-L-CD1595-GFP. (**B**) Viral replication kinetics of VSV-GFP, VSV-L-MT1620-mCherry, and VSV-L-CD1595-GFP compared in BHK-21 and HEp-2 cells at 33 °C and 37 °C. Titers were quantified using TCID_50_ assays at 33 °C, a permissive temperature for VSV-L-CD1595-GFP. Data shown as mean (SD) from two independent samples. (**C**) Viral replication kinetics of different VSV strains. Single-step growth kinetics of VSV-GFP, VSV-L-MT1620-mCherry, and VSV-L-CD1595-GFP in A549, 293T, and 3T3 cells at 33 °C and 37 °C. Titers were quantified using TCID_50_ assays. Data shown as mean (SD) from two independent samples. (**D**) Comparison of virus-induced cytotoxic activity in an IFN response thiazolyl blue tetrazolium bromide (MTT) viability assays. IFN responsive BHK-21 cells were treated with increasing amounts (10, 100, 500, and 1000 U/mL) of IFN for 16 h before infection with MOIs 0.1, 1, and 10. The viability is shown normalized to the untreated control. Bars represent means +/- SEM (*n* = 4).

**Figure 4 viruses-11-00989-f004:**
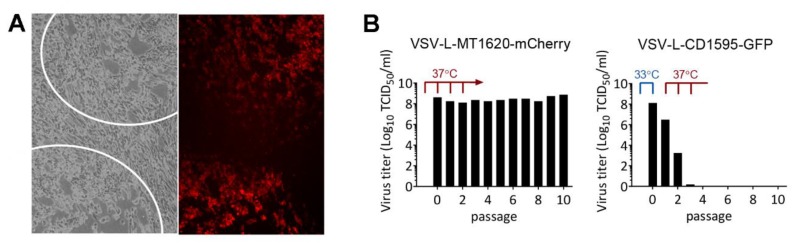
L-mCherry insertion viruses are genetically stable. Assessment of plaque fluorescence after passaging: (**A**) BHK-21 cells were infected with VSV-L-MT1620-mCherry in serial dilutions. Plaques were identified by cytopathic effect (CPE; delineated by circles) and then checked for fluorescence. (**B**) VSV-L-MT1620-mCherry and VSV-L-CD1595-GFP passages were performed at 37 °C (33 °C for initial L-CD1595-GFP variant production) and titrated with TCID_50_. Titrations were performed at the corresponding permissive temperatures (37 °C for L-MT160-mCherry variant; 33 °C for L-CD1595-GFP variant).

**Figure 5 viruses-11-00989-f005:**
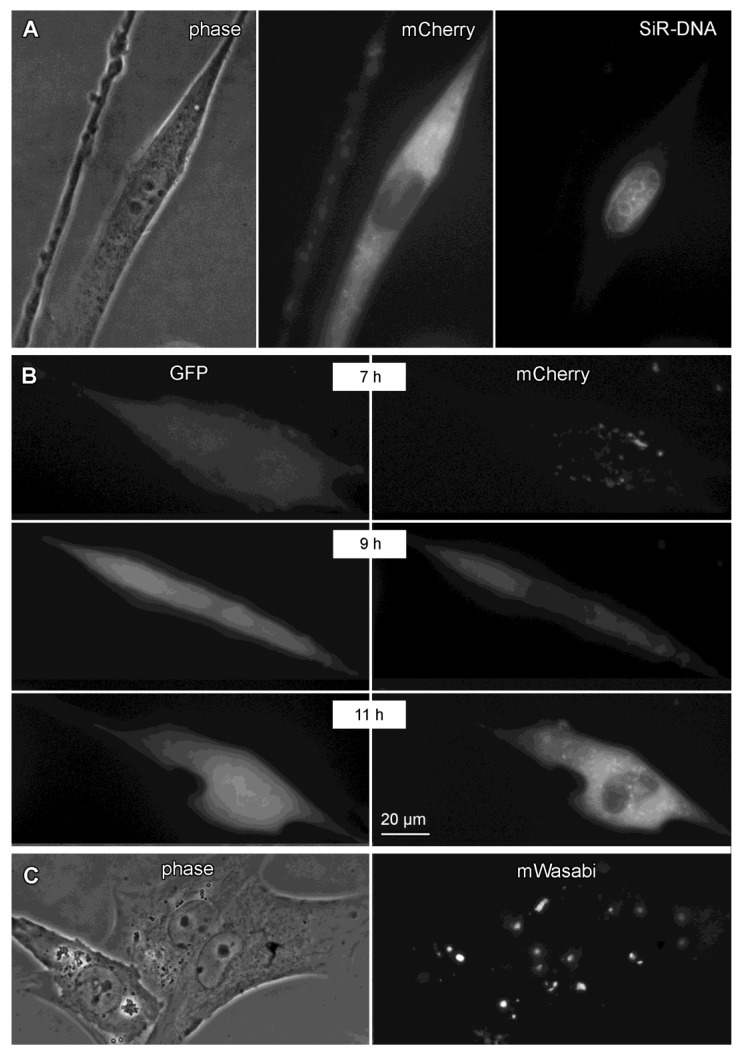
High magnification fluorescence microscopy shows distinctive fluorescence pattern of GFP, L-mCherry, and L-mWasabi. (**A**) Assessment of putative viral inclusion bodies: BHK-21 cells were infected with VSV-L-mCherry at MOI of 10. Images of one representative cell in phase contrast, tetramethylrhodamine (TRITC), and infrared (0.5 µM SiR-DNA for nuclear staining) channel are shown. (**B**) Assessment of the kinetic of GFP vs. mCherry fluorescence in time-lapse recording: BHK-21 cells were infected with VSV-GFP-L-mCherry at MOI of 10. Images were acquired for 12 h after infection. Images of one representative cell in fluorescein isothiocyanate (FITC) and TRITC channels at three time points are shown. (**C**) Assessment of putative viral inclusion bodies in a representative micrograph of BHK-21 cells infected with VSV-L-mWasabi at MOI of 10. Scale bar: 20 μm.

**Figure 6 viruses-11-00989-f006:**
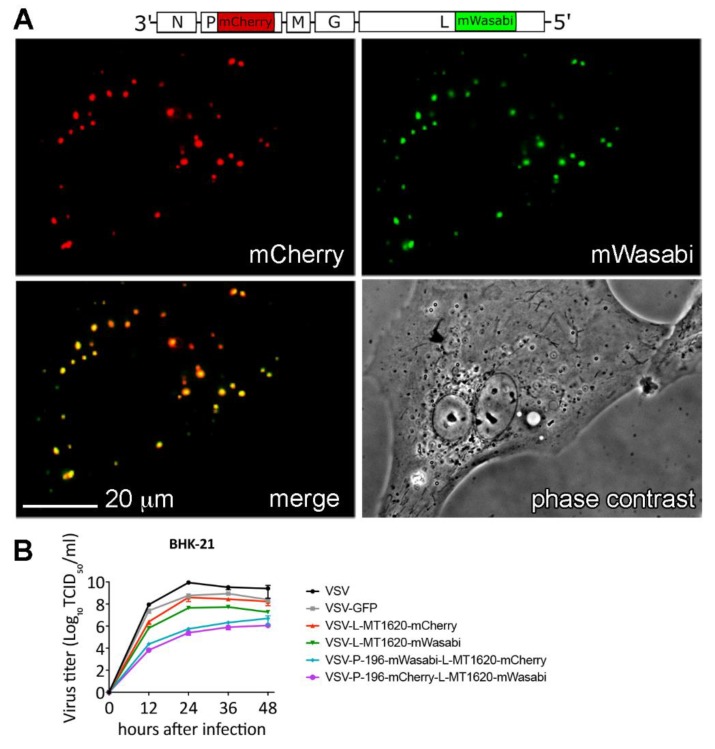
Attenuated double P- and L-insertion viruses display co-localizing fluorescent signals: (**A**) Assessment of putative P- and L-protein interaction in a representative micrograph of BHK-21 cells infected with VSV-P-mCherry-L-mWasabi at MOI of 10 at 10 h post infection. (**B**) Viral replication kinetics of parental VSV, VSV-GFP, VSV-L-MT1620-mCherry, VSV-L-MT1620-mWasabi, VSV-P-mWasabi-L-mCherry, and VSV-P-mCherry-L-mWasabi compared in BHK-21 at 37 °C in a multistep replication kinetics (MOI 0.1) at time points 12, 24, 36, and 48 h after infection. Titers were quantified using TCID50. Data shown as mean (SD) from two independent samples.

**Table 1 viruses-11-00989-t001:** L-expression plasmid cloning strategy primer. italics—L sequence; underline—mCherry sequence.

Name	For/Rev	Sequence (5′-3′ Direction)
CD1506insertGGSG for	for	GGCTCAGGCGGTGGATCCGGC*CTGTGGCTGTTCAGCGACG*
CD1506insertGGSG rev	rev	GCTCCCTCCGCCGCTTCCGCC*CTGGCTGTAGTGGCTGCGG*
CD1537insertGGSG for	for	GGCTCAGGCGGTGGATCCGGC*AGCGGCAAGGACAAGAACGAG*
CD1537insertGGSG rev	rev	GCTCCCTCCGCCGCTTCCGCC*CAGGAAGGGCTTGTACAGGATC*
MT1603insertGGSG for	for	GGCTCAGGCGGTGGATCCGGC*GAGAGCCGCGGCACCATCAC*
MT1603insertGGSG rev	rev	GCTCCCTCCGCCGCTTCCGCC*GCGGCCCCAGGGAGGGTAG*
MT1620insertGGSG for	for	GGCTCAGGCGGTGGATCCGGC*TACCCCAAGATGCTGGAGATGC*
MT1620insertGGSG rev	rev	GCTCCCTCCGCCGCTTCCGCC*TGGGGTGGTGGTGTAGTACAC*
MT1889insertGGSG for	for	GGCTCAGGCGGTGGATCCGGC*CAGTTCATCCCCGACCCCTTC*
MT1889insertGGSG rev	rev	GCTCCCTCCGCCGCTTCCGCC*GCTGGGGATGCCGGTCAGG*
mCherry-GGSG for	for	GGCGGAAGCGGCGGAGGGAGCGGGGGCGGGAGCGGAATGGTGAGCAAGGGCGAGG
mCherry-GGSG rev	rev	GCCGGATCCACCGCCTGAGCCGCCTCCGGACCCTCCCTTGTACAGCTCGTCCATG
Gibson new for	for	*GGCAGGGTCGGAACAGGAG*
Gibson new rev	rev	*CAGGCGTTTCCCCCTGGAAG*

**Table 2 viruses-11-00989-t002:** Vesicular stomatitis virus (VSV) plasmid cloning strategy primer. italics—L sequence; underline—mCherry sequence.

Name	For/Rev	Sequence (5′-3′ Direction)
CD1506insertGGSG rev	for	GGCTCAGGCGGTGGATCCGGC*CAATTATGGTTATTCTCAG*
CD1506insertGGSG for	rev	GCTCCCTCCGCCGCTTCCGCC*TGAATAATGTGATCTGTATTTTC*
MT1620insertGGSG rev	for	GGCTCAGGCGGTGGATCCGGC*TACCCAAAGATGCTAGAGATG*
MT1620insertGGSG for	rev	GCTCCCTCCGCCGCTTCCGCC*AGGGGTGGTCGTATAATAAAC*
mCherry-GGSG for	for	GGCGGAAGCGGCGGAGGGAGCGGGGGCGGGAGCGGAATGGTGAGCAAGGGCGAGG
mCherry-GGSG rev	rev	GCCGGATCCACCGCCTGAGCCGCCTCCGGACCCTCCCTTGTACAGCTCGTCCATG
49bp-before-FseI	for	*GCTGCCAAGTAATACACCGG*
50bp-after-SfoI	rev	*TTTATCTCCTCCTAAAGTTTC*
196-GGSG-P for	for	GGCTCAGGCGGTGGATCCGGC*GTTTGGTCTCTCTCAAAGACAT*
196-GGSG-P rev	rev	GCTCCCTCCGCCGCTTCCGCC*ATCTGATACTGCTTCTGATTGG*
33bp-before-Bst1107l	for	*AAGGAATGCCCGACAGCC*
35bp-after-XbaI	rev	*TCCGTCACCTCCGACAGAG*
CD1595insertGFP for	for	GGCATGGACGAGCTGTACAAG*ATGAGCTATCCCCCTTGGGG*
CD1595insertGFP rev	rev	CAGTTCCTCGCCCTTGCTCAT*GTCTTTATTATTATCCTTAGCAATCCCG*
GFP for	for	ATGAGCAAGGGCGAGGAACT
GFP rev	rev	CTTGTACAGCTCGTCCATGCC

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
