# Peer review of "The Methyltransferase Region of Vesicular Stomatitis Virus L Polymerase Is a Target Site for Functional Intramolecular Insertion"

_viruses, 2019, doi:10.3390/v11110989_

Round 1

Reviewer 1 Report

The authors take advantage of resent structural characterization of the VSV-L protein to identify sites for transgene insertion that result in stable VSV constructs with minimal attenuation characteristics. The authors in conclusion establish a system for the further characterization of VSV replication machinery. The recent success of this vector as a vaccine candidate pushes studies like this to the forefront, and are essential to developing new avenues of development by characterizing its ability to tolerate trangenes throughout the genome. The article was well written and presented and I would recommend it for acceptance.

Minor Comment: The authors use a full complement of VSV genes in their methods to rescue virus. It is this reviewers experience that the supplementation of M+G proteins are not needed when rescuing pseudotyped virus. A bit more explanation as too how the authors optimized their reverse genetics system in the methods would be helpful.

Author Response

We thank reviewer 1 for the appreciation of our study and the supportive feedback. Reviewer 1 had a minor comment on our methodology pertaining to the VSV rescue protocol.

From the reviewer:

Minor Comment: The authors use a full complement of VSV genes in their methods to rescue virus. It is this reviewers experience that the supplementation of M+G proteins are not needed when rescuing pseudotyped virus. A bit more explanation as too how the authors optimized their reverse genetics system in the methods would be helpful.

Authors response:

Reviewer 1 is correct in stating that M and G plasmids are optional in current rescue protocols of rhabdoviruses. While we do indeed apply such rescue protocols for some of our VSV generations, we revert to the classical rescue protocol in cases when challenges during rescue are expected. Such is the case in generation of chimeric VSV variants with G-protein exchange (such as our VSV-GP construct that we have been working on for many years). To add both M and G to the protocol in this current study was driven in part by the previously reported L-insertion virus variants that had been described to be strongly attenuated. We adapted the wording in line 183 – line 186:

Revised manuscript lines 183-186:

M and G proteins as helper plasmids are optional in the recovery of VSV, chosen here as a precaution to support the rescue of potentially attenuated virus variants. After the rescue, viruses were passaged on 293-VSV cells and plaque purified twice on BHK-21 cells.

Reviewer 2 Report

Heilmann and colleagues set out to generate a novel recombinant reporter VSV containing a fluorescent protein inserted in the L protein. They tested multiple locations, and found that insertion of a fluorophore at position 1620 can yield a replication competent reporter virus. They demonstrate that inserting a fluorescent protein in the VSV methyltransferase domain does not significantly affects its growth kinetics in vitro in various cell lines, though the presence of a fluorophore at any location in the genome affects the virus’ ability to overcome the IFN response.

Overall, this study is performed and written well. These reporter viruses are valuable for further studies into L protein function. The genome schemes are very helpful. This study can help guide reporter gene insertion in more distantly related viruses, which is especially valuable as inserting foreign genes in viral polymerases remains a challenge.

A few areas of this manuscript could be improved, including the order of some data, and the material and method section is somewhat incomplete. Furthermore, I would strongly encourage the inclusion of a growth kinetics comparison with unlabeled VSV.   

Major comments

In one of the last results sections, the authors describe secondary mutations found during (whole genome?) sequencing. Generally, it is good practice to fully sequence any newly generated recombinant mutant viruses to ensure that they contain the correct mutation(s), and only those. It’s a bit odd that the authors discuss the additional mutations at the end of the manuscript, instead of at the beginning. Whether these mutations have a phenotypic effect remains unclear. Authors state that no virus w/o these mutations could be generated, but it would appear that only 2 clones for each mutant were tested. This section should be moved up so it directly follows the section describing the initial rescue of these viruses. Furthermore, authors should indicate which viruses (containing which additional mutations) were used for the experiments shown in this manuscript. The fact that all these mutations are non-synonymous, makes me wonder whether there are also silent mutations present, that the authors don’t report. Generally, I feel that the order in which the data is presented is not necessarily the most logical one. It would make sense to move the section describing genomic stability up, placing it before the ‘high-magnification fluorescence microscopy’ section. It disrupts the flow of the paper somewhat that the Wasabi mutant is introduced twice, though not characterized. The difference in cellular staining is already noticeable in fig 2, but isn’t discussed until fig 4. The last section ‘a dual insertion VSV-P-mCherry-L-mWasabi’ feels incomplete. The material and methods section is lacking important details. It would also be helpful if the material and methods section reflects the order of the results section. It is unclear what imaging equipment was used (fluorescence microscopy and time-lapse exp). Are glass bottom dishes also used for the time-lapse experiment? The minigenome assay is described under ‘transfections’, which is confusing. A section describing sequencing of the viruses is missing In the virus recovery section, please describe what cell line was used to rescue viruses in. Generally, when characterizing a novel reporter virus, it is important to compare growth kinetics to its unlabeled parental virus, since insertion of foreign genes often results in growth attenuation. Authors compare growth kinetics of their novel reporter virus with the established GFP one, but not to an unlabeled/unmodified VSV. I do not suggest that the authors repeat all the growth curves currently in the manuscript, but I would strongly recommend repeating it in at least one key cell line while including an unlabeled virus. Figure 2A/Line 283: ‘However, only two sites (CD1506, MT1620) showed eGFP signal’ – some GFP signal is visible in the CD1537 panel (though I understand why authors did not want to move forward with that insertion site, considering the low levels of mCherry). Figure 2B: Some GFP+ cells do not appear to express mCherry, indicating loss of mCherry (or contamination of the GFP/mCherry stock with parental virus only containing GFP). Ideally, genome stability should be assessed for this mutant virus too. Figure 3C: The growth kinetics of the MT1620-mCherry virus in the 3T3 cells look a bit odd, plateau-ing between ~4-8hpi, before continuing upwards. At the 8hpi(?) timepoint the mCherry viruses have 2-log lower titers than VSV-GFP, and therefore the statement ‘No differences in titer were observed between incubations at 33C and 37C and titer differences between the virus variants only marginal’ is incorrect. How often was this growth curve repeated? In addition, please specify in the figure legend what the data in 3B-C represent (# of replicates and SD/SEM) Not requesting additional experiments, but growth kinetics performed at lower MOI (0.1-0.01) would have been valuable. The effect of IFN on growth kinetics cannot accurately be assessed using the current MOIs. Generally, it takes at least 24h before the effects of virus-induced IFN responses (and corresponding upregulation of ISGs) kick in. Figure 4: if authors used time-lapse to image these cells, why does it appear to be different cells at each timepoint?

Minor comments

Material/Methods section ‘Immunoblotting’: If you are only including the samples harvested 8hpi in the manuscript, it is not necessary to describe harvesting at 4, 8, 12 and 24hpi. Figure 1D: please indicate in the figure legend that mCherry is depicted in purple Line 278: please specify what the insertions were (Im assuming mCherry here?). In the material and methods section these plasmids are called ‘L-mCherry expression plasmids’. Consistency in nomenclature helps the reader. Figure 2: The title ‘Insertion of mCherry at position MT1620 leads to replication-competent virus’ suggests that mCherry is a requirement. Phrasing it along the line that insertion at this position allows/does not interfere with rescue of replication-competent virus would be better. Same at line 300. Figure 2A: the figure legend implies that mCherry and GFP are not imaged at the same time, and therefore not the same field is shown for mCherry/GFP? The MM suggests that both fluorophores were imaged at the same time (48hpi) Line 301: I would rephrase ‘eGFP positive clones’ to something better describing L expression plasmids containing mCherry insertions, that retain polymerase activity. In addition, you don’t ‘clone the sites into the VSV genome’, authors inserted the mCherry gene at the indicated sites Line 312-314: ‘To address whether the insertion site only tolerates mCherry or other reporter genes also, we cloned and VSV-L-MT1620-mWasabi’. The resulting recombinant construct showed green fluorescence and replicative capability comparable to the mCherry counterpart.’ – replicative capability cannot be accurately be assessed without performing growth curves. Authors may mean intact polymerase activity? Line 325-327: ‘We next compared MT1620 mCherry and the previously published temperature sensitive CD1595 GFP insertion variants with wildtype L VSV counterparts for their replication kinetics and cytotoxic effects on BHK and HEp-2 cells at 33C and 37C (Figure 3). – Authors refer back to Figure 3B, which does not show CPE data. The CPE data described in Fig 3D is only done on BHK cells at 37C. Line 335: it would be helpful if the main text described when the IFN was added (xh before/after infection). This information is also missing from the figure legend Line 368: ‘This variant displayed similar growth characteristics as the mCherry insertion mutant (data not shown).’ – If authors have this data, please include it (especially after making earlier claims of this mutant virus having ‘replicative ability’ kinetics) The sequences in Fig S1 appear mislabeled, as the mutations change places after passage.

Author Response

Reviewer #2:

We thank reviewer 2 for the very helpful suggestions to improve our study and manuscript. Although not explicitly requested, we did add some new data and also revised the manuscript at several sections. In the following we list our detailed response:

 From the reviewer:

Major comments:

In one of the last results sections, the authors describe secondary mutations found during (whole genome?) sequencing. Generally, it is good practice to fully sequence any newly generated recombinant mutant viruses to ensure that they contain the correct mutation(s), and only those. It’s a bit odd that the authors discuss the additional mutations at the end of the manuscript, instead of at the beginning. Whether these mutations have a phenotypic effect remains unclear. Authors state that no virus w/o these mutations could be generated, but it would appear that only 2 clones for each mutant were tested. This section should be moved up so it directly follows the section describing the initial rescue of these viruses. Furthermore, authors should indicate which viruses (containing which additional mutations) were used for the experiments shown in this manuscript. The fact that all these mutations are non-synonymous, makes me wonder whether there are also silent mutations present, that the authors don’t report.

Authors response:

We agree with the referee to reconsider the presentation of the description of mutations. Originally we omitted the important information that sequencing of all generated viruses revealed no silent mutations. We changed that. Mutations found were described in the paper. We also placed the sequencing paragraph further up as suggested. Not part of this paper, but we did observe the need for these additional mutations in a number of other VSV constructs with L-protein insertions of elements different to reporter genes, hence our generalized claim. We try to present this as a purely descriptive finding acknowledging that the data alone do not merit the claim for required mutations. Detail information on the mutation make up of each individual virus variant has been added to the manuscript.

Revised manuscript lines 468-479:

“Upon whole genome sequencing of rescued VSV-L-insertion variants we observed one to two secondary non-synonymous mutations in all viruses, which are located in proximity to the site of insertion (suppl. Figure S1 and S2). We did not find any silent mutations. The mutations in L-mWasabi viruses were located upstream (mWasabi 1: K1402R; mWasabi 2: R1410T) and downstream (mWasabi 1: M1936I; mWasabi 2: I1899L) of mWasabi. VSV-L-mWasabi with mutations K1402R and M1936I was used for further experiments (including for generation of a double insertion virus described below). The mutations in L-mCherry viruses were located downstream of mCherry (VSV-GFP-L-mCherry (mCherry 1): C2098G, E2107A; VSV-L-mCherry (mCherry 2): N2109Y). We were not able to rescue viruses that did not harbor associated mutations in the L protein. Whether these mutations are conditional and required for proper polymerase function remains to be studied.”

Generally, I feel that the order in which the data is presented is not necessarily the most logical one. It would make sense to move the section describing genomic stability up, placing it before the ‘high-magnification fluorescence microscopy’ section.

Authors response:

We agree with this notion and adapted the order of data presentation: initial virus rescue description -> section on sequence confirmation -> replication kinetics /virus activity -> genomic stability -> high magnification imaging.

It disrupts the flow of the paper somewhat that the Wasabi mutant is introduced twice, though not characterized. The difference in cellular staining is already noticeable in fig 2, but isn’t discussed until fig 4.

Authors response:

We appreciate this comment and added a new sentences on the different staining pattern in the main text describing figure 2.

Revised manuscript lines 457-459:

“Notably, mWasabi fluorescence yield is stronger than mCherry’s [26], which results in earlier and stronger focal fluorescence signals.”

The last section ‘a dual insertion VSV-P-mCherry-L-mWasabi’ feels incomplete.

Authors response:

The initial presentation of the dual reporter insertion virus was meant as a proof-of-concept. Such viruses could be used for more in-depth studies on interactions within the polymerase complex. However, as discussed below in section 6, we performed additional in vitro studies and included a multi-step growth kinetics of several viruses, including double insertion viruses, in a newly extended Figure 6.

Revised manuscript lines 594-598:

Both viruses showed colocalizing fluorescence signals in FITC and TRICT channels, putatively representing the interaction of P-mCherry and L-mWasabi fusion protein interactions (Figure 6A). Both double insertion viruses produce significantly lower titers compared to parental VSV (up to 5 logs at 12-24 hours), VSV-GFP and single L-insertion VSV variants (about 2 logs at 12-24 hours), as shown in a multi-step (MOI 0.1) replication kinetic at 37°C (Figure 6B). Data shown as mean (SD) from two independent samples.

The material and methods section is lacking important details. It would also be helpful if the material and methods section reflects the order of the results section.

Authors response:

We appreciate these comments and have therefore added details as well as revised the listed items (a-e) in the method section.

a) It is unclear what imaging equipment was used (fluorescence microscopy and time-lapse exp). b) Are glass bottom dishes also used for the time-lapse experiment?

Revised manuscript lines 251-261:

“Fluorescence microscopy and time lapse recordings

2*105 BHK-21 cells were seeded in polylysine-coated glass bottom dishes (ibidi GmbH, Gräfelfing, Germany) one day before infection. Nuclei were stained with 0.5 µM Sir-DNA (tebu-Bio GmbH, Offenbach, Germany) 30 minutes prior to infection and P-glycoproteins were inhibited with 5 µM verapamil for better Sir-DNA staining. BHK-21 cells were infected with an MOI of 10. Single images were acquired 6-10 hours after infection at 37° using a 63X/NA1.4 objective on an automated live cell imaging Zeiss Axiovert 200M microscopy equipped with a Sola light engine LED light source (Lumencor, Visitron Systems GmbH, Puchheim, Germany), a pco.edge 4.2 scMOS camera (PCO AG, Kelhaim, Germany), controlled by VisiView software (Visitron). Exposure times were 200 ms for mCherry/mWasabi, 50 ms for Sir-DNA and 10 ms for phase contrast. For time lapse recordings images were taken for 8 hours every 10 minutes from 10 different positions.”

c) The minigenome assay is described under ‘transfections’, which is confusing.

Revised manuscript line 172:

Header was changed to “Mini-genome assay”.

d) A section describing sequencing of the viruses is missing

A section describing virus sequencing reactions preparation was added.

Revised manuscript lines 212-218:

“Virus sequencing.

Genomic viral RNA of viruses generated in this study was purified with PeqGOLD Viral RNA kit and cDNA synthesis performed with RevertAid First Strand cDNA synthesis kit according to manufacturer’s recommendations. 10 overlapping PCR products per virus genome were generated. Subsequently, PCRs were column purified and Sanger sequenced by Microsynth AG with up to 40 different primers, i.e. sequencing reactions, depending on amount of inserts (1 or 2) and sequence quality, to obtain whole genome sequences.”

e) In the virus recovery section, please describe what cell line was used to rescue viruses in.

Revised manuscript Methods line 181:

…VSV-L-CD1595-GFP and VSV-P-mCherry-L-mWasabi were rescued in 293T cells…

also see minor comment, reviewer #1

Generally, when characterizing a novel reporter virus, it is important to compare growth kinetics to its unlabeled parental virus, since insertion of foreign genes often results in growth attenuation. Authors compare growth kinetics of their novel reporter virus with the established GFP one, but not to an unlabeled/unmodified VSV. I do not suggest that the authors repeat all the growth curves currently in the manuscript, but I would strongly recommend repeating it in at least one key cell line while including an unlabeled virus.

Authors response:

We appreciate the suggestion to also include unlabeled parental VSV in a growth kinetic experiment. We initially compared our insertion variants to VSV-GFP to normalize the presence of a fluorescence reporter gene.

However, we agree that a parental VSV comparison would be crucial to fully assess a potential attenuation and performed an additional study. We compared VSV, VSV-GFP. VSV-L-mCherry, VSV-L-mWasabi, VSV-P-mCherry-L-mWasabi, and VSV-P-mWasabi-L-mCherry via a multi-step kinetic with an MOI of 0.1, as suggested by the reviewer. These data are presented in the new FIGURE 6A.

Revised manuscript lines 594-598:

Both double insertion viruses produce significantly lower titers compared to parental VSV (up to 5 logs at 12-24 hours), VSV-GFP and single L-insertion VSV variants (about 2 logs at 12-24 hours), as shown in a multi-step (MOI 0.1) replication kinetic at 37°C (Figure 6B). Data shown as mean (SD) from two independent samples.

Figure 2A/Line 283: ‘However, only two sites (CD1506, MT1620) showed eGFP signal’ – some GFP signal is visible in the CD1537 panel (though I understand why authors did not want to move forward with that insertion site, considering the low levels of mCherry). Figure 2B: Some GFP+ cells do not appear to express mCherry, indicating loss of mCherry (or contamination of the GFP/mCherry stock with parental virus only containing GFP). Ideally, genome stability should be assessed for this mutant virus too.

Authors response:

Delta-L VSV-GFP was produced on 293-VSV cells, expressing N, P-GFP and L. This results in viruses carrying functioning L-protein and GFP labelled P. Although infection with low MOI allows a clear distinction between properly complemented viruses and variants with only residual activity, some GFP leakiness still occurs due to remaining wt L and P-GFP.

Regarding the dual fluorescence image in 2B: the signal intensity in the L-inserted mCherry is significantly lower than the cytosolic GFP signal from a separate transgene. This results some cells displaying very low mCherry intensities. We slightly modified Fig 2B by digitally enhancing the exposure time (linear brightness correction in photoshop, no non-linear modification was applied). This results in a closer match of the two fluorescent reporters.

Figure 3C: The growth kinetics of the MT1620-mCherry virus in the 3T3 cells look a bit odd, plateau-ing between ~4-8hpi, before continuing upwards. At the 8hpi(?) timepoint the mCherry viruses have 2-log lower titers than VSV-GFP, and therefore the statement ‘No differences in titer were observed between incubations at 33C and 37C and titer differences between the virus variants only marginal’ is incorrect. How often was this growth curve repeated? In addition, please specify in the figure legend what the data in 3B-C represent (# of replicates and SD/SEM)

Authors response:

We agree that 3T3 replication kinetics look unusual. We therefore repeated the experiment with a more vigorous washing step. The repeated growth kinetics curve is much improved and we revised Figure 3C. Hence we attribute the initial unusual kinetic curve for the insert variant at the 8 hour timepoint to a technical error.

All growth kinetic experiments were performed in duplicates from two separate culture wells. Serial dilution and TCID50 titration were performed independently on each sample. The data are presented as mean (SD) from those duplicates.

Revised manuscript lines 523, 526 and 607:

Data shown as mean (SD) from two independent samples.

Not requesting additional experiments, but growth kinetics performed at lower MOI (0.1-0.01) would have been valuable. The effect of IFN on growth kinetics cannot accurately be assessed using the current MOIs. Generally, it takes at least 24h before the effects of virus-induced IFN responses (and corresponding upregulation of ISGs) kick in.

Authors response:

We opted for single step growth curves in the initial draft to solidly compare the single cycle replication capacity of the newly generated viral variants. However, we performed an additional comparative growth kinetic study (also addressing points #4 and #6) and this study was performed as multistep growth curve using an MOI of 0.1. While we did not test for differences in the induction of IFN and ISGs between the newly generated virus variants, our MTT killing assays with varying doses of IFN and MOIs ranging from 0.1 to 10 shows an increased IFN sensitivity at the lower MOI’s. To highlight that this study was performed after IFN pre-incubation of 16 hrs, we added this information to the figure legend

Revised manuscript lines 594-598:

Both double insertion viruses produce significantly lower titers compared to parental VSV (up to 5 logs at 12-24 hours), VSV-GFP and single L-insertion VSV variants (about 2 logs at 12-24 hours), as shown in a multi-step (MOI 0.1) replication kinetic at 37°C (Figure 6B). Data shown as mean (SD) from two independent samples.

Revised manuscript line 529:

of IFN for 16 hours before infection with MOIs

Figure 4: if authors used time-lapse to image these cells, why does it appear to be different cells at each timepoint?

Authors response:

Time lapse cell images in FIGURE 4 show the same cell but with long time intervals in between (2 hours each). Consequently, cell morphology changed particularly in light of the infection. The example images were originally from an extended time lapse recording. We generated movies from the time lapse data, which we now include as a supplemental file. In these movies, the morphological transitions are depicted.

Revised manuscript line 564:

using time-lapse analysis (Figure 5B, supplemental Video 1)

File description for time laps video file:

File S3. High magnification fluorescence microscopy movie underlines distinctive fluorescence pattern of GFP and L-mCherry. Assessment of the kinetic of GFP vs. mCherry fluorescence in time-lapse recording: BHK-21 cells were infected with VSV-GFP-L-mCherry at MOI of 10 at 37°C. Images were acquired for 12 hours after infection. A movie was generated from 10-minute interval images of one representative cell in phase contrast (left), TRITC (middle) and FITC (right) channels.

Minor comments:

Material/Methods section ‘Immunoblotting’: If you are only including the samples harvested 8hpi in the manuscript, it is not necessary to describe harvesting at 4, 8, 12 and 24hpi.

We corrected the respective section.

Revised manuscript line 191:

…cell lysates were prepared 8 hours later.

Figure 1D: please indicate in the figure legend that mCherry is depicted in purple

Revised manuscript line 390:

Molecular model of VSV L-protein with mCherry (depicted in purple) insertion at position MT1620.

Line 278: please specify what the insertions were (Im assuming mCherry here?).

In the material and methods section these plasmids are called ‘L-mCherry expression plasmids’. Consistency in nomenclature helps the reader.

Revised manuscript line 398:

L-mCherry expression vector plasmids with insertions at CD1506, CD1537, MT1603,

Figure 2: The title ‘Insertion of mCherry at position MT1620 leads to replication-competent virus’ suggests that mCherry is a requirement. Phrasing it along the line that insertion at this position allows/does not interfere with rescue of replication-competent virus would be better. Same at line 300.

Revised manuscript line 410:

Figure 2. Insertion of mCherry at position MT1620 is compatible with VSV replication.

Revised manuscript line 421:

MT1620 insertion is compatible with VSV replication.

Figure 2A: the figure legend implies that mCherry and GFP are not imaged at the same time, and therefore not the same field is shown for mCherry/GFP? The MM suggests that both fluorophores were imaged at the same time (48hpi)

The images were indeed taken at the same time from the same micrograph frame. We added this information to the figure legend.

Revised manuscript lines 414-415:

Bottom: GFP signal after inoculation with VSV-GFP-ΔL at an MOI of 10 depicted from the same micrograph frames as above.

Line 301: I would rephrase ‘eGFP positive clones’ to something better describing L expression plasmids containing mCherry insertions, that retain polymerase activity. In addition, you don’t ‘clone the sites into the VSV genome’, authors inserted the mCherry gene at the indicated sites

We corrected the respective section.

Revised manuscript lines 422-423:

Next we cloned the mCherry gene into the full-length VSV genome at L-protein sites CD1506 and MT1620. Insertions were performed into two VSV backbones,

Line 312-314: ‘To address whether the insertion site only tolerates mCherry or other reporter genes also, we cloned and VSV-L-MT1620-mWasabi’. The resulting recombinant construct showed green fluorescence and replicative capability comparable to the mCherry counterpart.’ – replicative capability cannot be accurately be assessed without performing growth curves. Authors may mean intact polymerase activity?

We appreciate this correction and have replaced “replicative capability” with “intact polymerase activity” to reflect the initial observations. At the subsequent figure 6B we now provide new data for replicative capability of the mWasabi insert variants.

Revised manuscript line 457:

… and intact polymerase activity comparable to the mCherry counterpart.

Line 325-327: ‘We next compared MT1620 mCherry and the previously published temperature sensitive CD1595 GFP insertion variants with wildtype L VSV counterparts for their replication kinetics and cytotoxic effects on BHK and HEp-2 cells at 33C and 37C (Figure 3). – Authors refer back to Figure 3B, which does not show CPE data. The CPE data described in Fig 3D is only done on BHK cells at 37C.

We modified the corresponding section.

Revised manuscript lines 498-500:

“We next compared MT1620 mCherry and the previously published temperature sensitive CD1595 GFP insertion variants with wildtype L VSV counterparts for their replication kinetics and cytotoxic effects on BHK and HEp-2 cells at 33°C and 37°C (Figure 3).”

Line 335: it would be helpful if the main text described when the IFN was added (xh before/after infection). This information is also missing from the figure legend

Revised manuscript line 509:

IFN was added 16 hours before infection.

Revised manuscript line 529:

…of IFN for 16 hours before infection with MOIs 0.1, 1 and 10.”

Line 368: ‘This variant displayed similar growth characteristics as the mCherry insertion mutant (data not shown).’ – If authors have this data, please include it (especially after making earlier claims of this mutant virus having ‘replicative ability’ kinetics)

We did not have a detailed growth kinetic before but virus yield, emergence of fluorescence and CPEs were comparable to mCherry variants. However, as discussed above, we now provide comparative replication kinetic in a new Fig 6B. We show marginal differences between mCherry or mWasaby insertion variants.

The sequences in Fig S1 appear mislabeled, as the mutations change places after passage. 

We corrected this mistake in a revised Figure S1.

Round 2

Reviewer 2 Report

The authors have clearly made an effort to address all my comments, and did so in a satisfactory manner